# Mental health dynamics of adolescents: A one-year longitudinal study in Harari, eastern Ethiopia

Gari Hunduma[1]*, Yadeta Dessie[2⊙], Biftu Geda[3], Tesfaye Assebe Yadeta[1], Negussie Deyessa[4]

1 School of Nursing and Midwifery, College of Health and Medical Sciences, Haramaya University, Harar, Ethiopia, 2 School of Public Health, College of Health and Medical Sciences, Haramaya University, Harar, Ethiopia, 3 Department of Nursing, College of Health Sciences, Madda Walabu University, Shashamene Campus, Shashamene, Ethiopia, 4 Department of Preventive Medicine, School of Public Health, College of Health Sciences, Addis Ababa University, Addis Ababa, Ethiopia

⊙ These authors contributed equally to this work.

* garihunduma@gmail.com

**Data Availability Statement:** All relevant data are within the manuscript and its Supporting Information files.

## Abstract

### Aims

This study aims to assess the dynamics of in-school adolescents' mental health problems in Harari regional state, eastern Ethiopia for a year.

### Materials and methods

Using multistage sampling technique, we conducted a year-long longitudinal study at three public high schools between March 2020 and 2021. Three hundred fifty-eight in-school adolescents were chosen by systematic random sampling for the baseline assessment, and 328 completed the follow-up assessment. We used self-administered, adolescent version of SDQ-25 Questionnaire to collect the data. Wilcoxon matched-pairs signed-rank test and McNemara's Chi-squared tests were used to examine the median difference and distribution of mental health problems between times one and two. Random-effects logistic regressions on panel data was used to identify factors associated with mental health problems. A p-value < 0.05 was considered as statistically significant.

### Results

The magnitude of overall mental health problems at baseline assessment was 20.11% (95% CI: 16–25), with internalizing problems accounting for 27.14% (95% CI: 23–32) and externalizing problems accounting for 7.01% (95% CI: 4.6–10.3). At the follow-up assessment, these proportions rose to 22.56% (95% CI, 18–27) for overall problems and 10.3% (95% CI, 7.7–14.45) for externalizing problems. On other hand, internalizing problems decreased unexpectedly to 22.86% (95% CI, 18.6–27.7) at follow-up assessment. Internalizing problem scores at time two were significantly lower than baseline among older adolescents, girls and those with average wealth index in our study cohort.

**Funding:** Haramaya University provided financial support for our study under the Scientific Research Grant number "HURG-2020-02-01-92". The financial support was fifty thousand Ethiopian birr. The funders had no role in study design, data collection and analysis, decision to publish, or preparation of the manuscript.

**Competing interests:** The authors have declared that no competing interests exist.

## Conclusions

The prevalence of mental health problems were high among the study cohort. The proportion of overall problems and externalizing problems has increased over time, indicating a deterioration in the mental health of the study cohort. However, the decrease in internalizing problems among older adolescents, girls, and those with an average wealth index is a positive sign. The findings highlight that tailored interventions are required to reduce externalizing problems and maintain the decrease in internalizing problems. These interventions should target middle-aged and male adolescents from low-income families.

## Introduction

Adolescence is a vulnerable time for mental illness and its effects due to the rapid social, psychological, cognitive, and emotional changes [1, 2]. Studies carried out in developed countries show that adolescent mental health problems, which can present as internalizing or externalizing mental health problems, have sharply increased over the past few decades [3, 4]. Internalizing (depression and anxiety) and externalizing (conduct disorders and hyperactivity) problems are adolescents' most commonly reported mental health problems [5]. Although nearly half of mental health disorders begin by age 14 and 75% start by age 24, the majority are underdiagnosed and undertreated [6, 7] or start much later [6]. Thus, failure to detect and treat the problem at an early stage can have detrimental long-term effects, including poor academic performance, unemployment, homelessness, poverty, various health problems, and involvement with the juvenile justice system [8–10].

One in seven (10–19 year old) people worldwide suffer from mental health disorders [11] which account for 16% of the worldwide burden of disease and disability [12]. According to a systematic analysis published in 2019, the global prevalence of common mental disorders among adolescents aged 10 to 19 years was 25% and 31.0%, using the GHQ cut-off point of 4 and 3, respectively [13]. It is estimated that 14.3% of adolescents in Sub-Saharan Africa suffer from mental health problems, with 10% having diagnosable psychiatric disorders [14].

A study of Ghanaian high school students found that the overall rate of mental health problems was 58.5%, with internalizing problems accounting for 36.8% and externalizing problems accounting for 16.3% [15]. According to their parents and adolescents, 31.2% and 40.8% of Tanzanian in-school adolescents had mental health problems, respectively [16]. In Uganda, Kinyanda et al. found an identically high self-reported prevalence of 41.4% [17]. Another study in central Kenya found that internalizing and externalizing problems were present in 17.2% and 5.4% of adolescents, respectively [18]. Few studies in Ethiopia have revealed that more than one-third of college or university students have common mental disorders [19]. Furthermore, the prevalence of childhood mental health problems is 12–25%, making it the country's health sector's highest burden of mental illnesses [20]. However, because these studies were all cross-sectional, they could not reveal the dynamics underlying adolescent mental health problems.

Adolescents' mental health problems persist into adulthood, impairing physical and psychological health and limiting opportunities to lead fulfilling lives as adults if not treated promptly [11, 21]. The problems can have long-term and short-term effects on employment, relationships, violence, substance abuse, reproductive and sexual health, educational attainment, and suicide [22]. Each of these factors can then impact the mental health of the next

generation. Furthermore, developing mental health problems during adolescence can contribute to high-risk behaviors such as self-harm, tobacco, alcohol and other substance use, and risky sexual behavior [12, 23, 24]. It also exposes adolescents to social exclusion, domestic violence, school dropout, discrimination, stigma, delinquency, and human rights violations [11].

Existing data show that various life factors, including social, demographic, and socioeconomic factors, influence adolescent mental health [25–28]. Exposure to adversity, peer pressure, identity exploration [11, 29, 30] and quality of home life are also factors that can contribute to mental health problems during adolescence [11, 31, 32]. Furthermore, violence (particularly sexual violence) and bullying at school endanger adolescents' mental health [11, 33, 34].

Early detection and intervention have improved adolescents' mental health, lowering delinquency, substance misuse, health-risking sexual behaviors, and school failure [10]. Understanding the dynamics of in-school adolescent mental health problems over time and the factors contributing to them is critical for designing effective interventions. However, there is still a scarcity of research on how adolescent mental health changes over time in developing countries like Ethiopia. Furthermore, no longitudinal studies have been conducted in Ethiopia to determine the stability of these issues over time. As a result, this study looked at the dynamics of in-school adolescents' mental health problems over a year. Therefore, this research aims to assess in-school adolescents' mental health dynamics in Harari Regional State, eastern Ethiopia, over a year. We hypothesized that the number of in-school adolescents with mental health problems would remain stable throughout the study, and therefore there would be no significant difference between the SDQ score of T1 and T2.

The finding may help to provide crucial data for policymakers in designing, planning, and implementing interventions of risk reduction and scale-up of services for adolescent mental health conditions.

## Methods and procedures

### Study setting, designs, and samples

A school-based longitudinal study design was conducted in three randomly selected high schools in Harari Regional State from March 2020 to March 2021. Harari region is located in eastern Ethiopia at 511 km from Addis Ababa. Unlike most other Ethiopian regions, the region's population is concentrated in urban areas (54.2%) [35]. Psychoactive substances like *khat* (Catha edulis), tobacco, and coffee play a significant role in the region's commercial operations. In addition, *khat* use is very common in the neighborhood, with around a quarter of the young people doing it [35]. In most of Hariri's rural sub-districts, *khat* is the main cash crop [36, 37].

The study was conducted over two round data collection periods, from March 1 to 16, 2020 (T1) and March 1 to 20, 2021 (T2). Participants in the study ranged in age from 14 to 19 and were enrolled in public high schools in the Harari regional state of eastern Ethiopia. During the data collection period, seven governmental high schools were in the region. We used a simple random sampling technique to select the schools. We listed all schools and randomly selected three schools using computer-generated sampling. We identified grade levels for each selected school, and finally, sections were selected proportionally using the lottery method from each grade level, considering the number of sections. Finally, participants were selected randomly from each section using systematic random sampling.

The baseline assessment was conducted in March 2020 with 358 randomly selected in-school adolescents from randomly selected schools. We collected data from the same participants for the second time in March 2021. In order to ensure that the data collected from

participants at different times could be linked, we used a unique ID code to identify participants for T2 data collection after a year. This allowed us to reconcile the data with T1 data analysis.

Three hundred twenty-eight (91.6%) of the 358 in-school adolescents who participated in baseline assessments completed the second assessment phase. We used the same technique and assessment tools for the follow-up assessment at school. The prevalence of mental health problems during the first phase of data collection was used as a baseline assessment. On the other hand, the prevalence of mental health problems in the second phase was used as a one-year follow-up to see if there was any significant change in problem dynamics. The current analysis included adolescents who completed baseline and follow-up assessments after one year.

## Data collection tool

We collected data from adolescent students at their schools using a guided, self-administered data collection technique. We used structured, standardized, self-administered questionnaires to collect the data. All data collection instruments were pre-tested in Dire Dawa administrative counselling among similar in-school adolescents in 5% of the sample size. We also calculated Cronbach's alpha for the reliability and validity of the tool to assess the scales' internal consistency and reliability before actual data collection.

Accordingly, the calculated Cronbach's alphas for the SDQ total, externalizing and internalizing, were 0.72, 0.63, and 0.66, respectively. Eight health professionals collected the data. They provided a suitable classroom for participant students in small groups of no more than 20 per session, as well as instructions on completing the questionnaires. Each session had two data collectors whose main role was to facilitate, read the questions to the participants if necessary and then let them choose their acceptable responses based on the questionnaire frameworks. The trained supervisors and the primary investigator closely monitored the data collection process.

## Variables and measurements

Mental health problems were dependent variables measured using the strength and difficulties questionnaire (SDQ-25). Socio-demographic variables (age, sex, residence); socio-economic status (education, occupation, and wealth of the family); psychosocial variables (bullying at school, Fear of COVID-19, risk of COVID-19 exposure); behavioral variables (substance use); biological factors (family history of mental illness, known chronic medical problems) are independent variables and were measured by using a standard questionnaire adopted from previous studies.

Mental health problems were assessed using the self-administered Strength and Difficulties Questionnaire (SDQ) [38–40]. The SDQ consists of 25 items divided into five subscales, with a fifth subscale used to determine strengths. Four subscales contribute to the overall difficulty score. Higher scores indicate a greater risk of developing mental health problems. A 3-point Likert-type scale is used in this survey (0 being not true, 1 being somewhat true, and 2 being true). Using the score banding technique outlined by Goodman, the SDQ total difficulties score is broken down into normal (0–13), borderline (14–16), and abnormal scores [41, 42].

We used the sum of the first four problems domains (excluding the prosocial behavior items), to generate a total difficulties score ranging from 0 to 40. The sum of conduct and hyperactivity scales was used to generate externalizing score which ranging from 0 to 20 and the sum of emotional and peer problem scales to generate internalizing score which ranging 0 to 20. The internalizing problem subscale category is normal (0–7), borderline (8) and

abnormal (9–20) while the externalizing subscale for normal, borderline and abnormal are (0–8), (9), (10–20) respectively [38, 43]. A borderline category score was considered a cut-off point for each difficulty sub-scores for indicating mental health problems in this study. The SDQ can be used within community samples to separate internalizing and externalizing problems. We looked at the internalizing and externalizing sub-scores and the discrete sum scores for the overall SDQ score in this study.

Cronbach's alpha is a measure of the internal consistency or reliability of a set of survey items. It quantifies the level of agreement on a standardized 0 to 1 scale, with higher values indicating higher agreement between items. Cronbach's alpha is used to determine whether a collection of items consistently measures the same characteristic. In this study, we assessed Cronbach's alpha to evaluate the internal consistency of our survey items. The Cronbach's α for SDQ total is 0.764 while it is 0.55 and .65 for internalizing and externalizing in the current sample respectively.

Fear of COVID-19 Scale (FCV-19S) was used to assess the severity of fear related to COVID-19 and has already been validated in many languages. It is a uni-dimensional scale that assesses the fear of COVID-19 and was developed by Ahorsu et al.2020 [44]. The instrument has seven items with a 5-point Likert scale (strongly disagree = 1) to (strongly agree = 5). The minimum possible score for each question is one, and the maximum is seven. The score obtained vary from 7 to 35, with higher scores indicating greater fear of COVID-19.

We assessed the risk of covid-19 by asking, "Is there anyone in your family and/or neighbors who are infected with covid-19?" We provided "yes" or "no" options, and if a participant chose "yes" for the first or second, or both, the participant was considered to have a risk of exposure to COVID-19.

Wealth index is a composite measure of the cumulative living standard of a household. It is calculated using data on a household's ownership of selected set of assets. In this case, the wealth index was measured using the number and kind of known goods (such as televisions, bicycles, and cars; dwelling characteristics such as flooring material; type of drinking water source; and toilet and sanitation facilities) the family owns as reported by the adolescent and analyzed using the principal component analysis [45].

Substance use was defined as the ever-current use of substances such as alcohol, cigarettes, khat, or other illicit drugs. To assess the use of substances among adolescents, we adapted and used a questionnaire from the Global School-Based Health Survey (GSHS), which is recommended by the World Health Organization (WHO). This questionnaire assesses adolescents' substances (including alcohol, cigarette and others) and their frequency of use in the previous 12 months [46].

Bullying at school was assessed whether a respondent has been bullied at school, they were asked the question "How often have you been bullied at school in the past couple of months?". If the answer was more than once a week, then the respondent was categorized as "1" for being bullied at school. Otherwise, the respondent was categorized as "0".

Self-esteem: defined as the "judgment one makes about their self-concept or "attitude one holds toward themselves as an object that measured via assessing a subject's attitude about themselves as a "thing." The 10-item Rosenberg Self-Esteem Scale is used to measure global self-esteem, which consists of statements related to feelings of self-worth and self-acceptance [47]. This 10-items scale ranges from (0 = strongly agree) to 3 (strongly disagree). The sum scores for all 10 items range from to 0–30 with higher scores indicating higher self-esteem. Respondents with a total score >25 were classified as having 'high self-esteem', scores between 15 and 25 were within the 'normal self-esteem' range, and scores below 15 suggested 'low self-esteem'.

## Data analysis

Data were entered into Epi data and analyzed by STAT 16 with a 95% confidence level for all statistical significance tests. Descriptive statistics, including mean, standard deviations, and percentages, were performed to characterize the sample regarding socio-demographic characteristics and outcome variables. Wilcoxon matched-pairs signed-rank test and McNemar's test Chi-squared tests were used to compare the median score severity charges of mental health problems among different groups of study participants. To find the potential factors associated with individual mental health disorders, we used random-effects logistic regressions on panel data. The dependent variable in each model was each mental health dimension's state (binary variable). We performed univariate analysis to check for the homoscedasticity and linearity of the variables before feeding them into the model. We ensured that the assumptions of linearity, independence, homoscedasticity, and normality were met before feeding the variables into the regression model. Statistical criteria of $P < 0.2$ during univariate analysis were used as a guide to enter independent variables into a multivariate model. Statistical significance was declared at ($P < 0.05$).

## Ethical considerations

Haramaya University's Institutional Health Research Ethics Review Committee (IHRERC) granted ethical approval with reference number IHRERC/149.2019. Participants were informed that participation in the study was entirely voluntary, that declining to participate would have no negative consequences for them or their families, and that they could stop at any time or skip any questions they did not wish to answer. Respondents got clear and adequate information concerning the research, including purpose, procedures, potential risks and benefits, and their right to participate in the study. Both participants and their parents informed that the information gathered would be disseminated to assist in knowledge generation only. For participants aged 13 to 17, written, informed, and signed voluntary consent was obtained from one of the parents or guardians, as well as written and signed voluntary consent from the adolescents. Participants aged 18 and up provided written, informed, and signed consent. The questionnaires did not include personal identifiers to ensure participant confidentiality. The study adhered to the Declaration of Helsinki's ethical principles for human subjects' medical research.

## Results

### Socio-demographic characteristics of participants

Thirty (8.4%) of the 358 in-school adolescents who took the baseline assessment did not receive follow-up assessments. Attrition was primarily due to changing schools and moving to different regions. As a result, 328 in-school adolescents were included in the final analysis, with a response rate of 94.6%. The respondents' mean age was 17.5 (SD 1.2), ranging from 14 to 20 years. About half the participants (50.9%) were male, and 62.5% primarily lived in urban areas. The majority (86.9%) of the participants were single, and 13.1% were engaged. About 32.9% of the participating adolescents lived with one parent (father or mother). Regarding the physical health of the adolescents, 18.6% experienced at least one chronic illness. About one-fifth (23.2%) of the respondents reported being bullied in school at least once a week during the last two months.

At the time of the data collection, one-eighth of the participants (12.2%) used cigarettes, one-seventh (14.3%) used alcohol, and one-third (32.3%) used *khat*. More than half of the participants (55.2%) expressed an intense fear of COVID-19, and 18.0% said they were at risk of

**Table 1. Characteristics of matched respondents in baseline survey (T1) and after one year (T2), (n = 328).**

| Characteristics | Categories | Baseline survey(T1) | After one year (T2) | p-Value** |
|---|---|---|---|---|
| **Age,** | Mean ± SD | 16.9 (1.3) | 17.5(1.2) | 0.001 |
| **Gender** | Male | 167 (50.9%) | 167 (50.9%) | - |
| | Female | 161 (49.1%) | 161 (49.1%) | |
| **Residence** | Urban | 205 (62.5) | 205 (62.5) | - |
| | Rural | 123 (37.5) | 123 (37.5) | |
| **Marital status** | Single | 292 (89.0) | 285 (86.9) | 0.001 |
| | Engaged | 36 (11.0%) | 43 (13.1%) | |
| **Any chronic disease** | No | 246 (75.0%) | 267 (81.4) | 0.001 |
| | Yes | 82 (25.0%) | 61 (18.6%) | |
| **Current living status** | Live with both parents | 220 (67.1%) | 220 (67.1) | - |
| | Live with one parent | 108 (32.9%) | 108 (32.9) | |
| **Mother educational status** | Not educated | 264 (80.49) | 264 (80.49) | - |
| | Educated | 64 (19.5) | 64 (19.5) | |
| **Father educational status** | Not education | 218 (66.5) | 218 (66.5) | - |
| | Educated | 110 (33.5) | 110 (33.5) | |
| **History of family mental illness** | No | 285 (86.9) | 285 (86.9) | - |
| | Yes | 43 (13.1) | 43 (13.1) | |
| **Wealth index** | Lowest | 134 (40.9%) | 131 (39.9) | 0.001 |
| | Middle | 131(39.9%) | 132 (40.2) | |
| | Highest | 63 (19.2%) | 65 (19.8) | |
| **Khat use** | No | 213 (64.9%) | 222 (67.7%) | 0.001 |
| | Yes | 115 (35.1%) | 106 (32.3%) | |
| **Alcohol use** | No | 265 (80.8%) | 281 (85.7%) | 0.001 |
| | Yes | 63 (19.2%) | 47 (14.3%) | |
| **Cigarette use** | No | 278 (84.8%) | 288 (87.8%) | 0.001 |
| | Yes | 50 (15.2%) | 40 (12.2%) | |
| **Self-esteem** | Low | 37 (11.3%) | 41 (12.5%) | 0.001 |
| | Normal | 213 (64.9%) | 200 (61.0%) | |
| | High | 78 (23.9%) | 87 (26.5%) | |
| **Suicide behaviors** | No | 288 (87.8%) | 289 (88.1%) | 0.001 |
| | Yes | 40 (12.2%) | 39 (11.9%) | |
| **Bullying at school** | No | 239 (72.9) | 252 (76.8) | 0.001 |
| | Yes | 89 (27.1) | 76 (23.2) | |
| **FCV-19S** | Low | NA | 147 (44.8%) | - |
| | High | NA | 181 (55.2%) | |
| **Risk of exposure to COVID-19** | No | NA | 269 (82.0%) | - |
| | Yes | NA | 59 (18.0%) | |

contracting the virus. Regarding parental characteristics, 66.5% of fathers and 80.5% of mothers were uneducated, while 33.5% of fathers and 19.5% of mothers were. Looking at the family history of mental problems, approximately (13.1) reported that one of their family members had a mental illness (Table 1).

## Dynamics of mental health problems for the year-long follow-up

The magnitude of mental health problems among adolescents was 20.11% (95% CI, 16–25) for overall mental health problems, 27.14% (95% CI, 23–32) for internalizing problems, and 7.01% (95% CI, 4.6–10.3) for externalizing problems at the baseline assessment (T1). At the

**Table 2. Pattern and prevalence of the self-reported adolescents' mental health problems with the corresponding time of the survey (T1 & T2): (n = 328).**

| Mental health problems domain | SDQ-25 profile with its corresponding time of the survey | | | | | |
|---|---|---|---|---|---|---|
| | Norma | | Borderline | | Abnormal | |
| | T1 | T2 | T1 | T2 | T1 | T2 |
| Internalizing problems N, % | 239 (72.9) | 253 (77.1) | 34 (10.4) | 31 (9.5) | 55 (16.8) | 44 (13.4) |
| Externalizing problems N, % | 305 (93.0) | 293 (89.3) | 8 (2.4) | 11 (3.4) | 15 (4.6) | 24 (7.3) |
| SDQ Total difficulties N, % | 262 (79.9) | 254 (77.4) | 29 (8.84) | 32 (9.8) | 37 (11.28) | 42 (12.8) |

The data showed that the difficulty score for overall, internalizing, and externalizing issues decreases as adolescents get older. Table 3 shows the observed change for each item as a function of age.

follow-up assessment (T2), this proportion increased to 22.56% (95% CI, 18–27) in overall problems and 10.3% (95% CI, 7.7–14.5) in externalizing problems but decreased to 22.86% (95% CI, 18.6–27.7) in internalizing problems (Table 2).

There was no statistically significant difference in the distribution of the severity of mental health problems from year to year. At T1, one-fifth (20.1%) of adolescents tested positive for

**Table 3. Prevalence distributions of self-reported adolescents' mental health problem symptoms (items) with ages and the corresponding time of the survey (t1 & t2) (N = 328).**

| Items | At age 14/15 | | At age 15/16 | | At age 16/17 | | At age 17/18 | | At age 18/19 | | At age 19/20 | |
|---|---|---|---|---|---|---|---|---|---|---|---|---|
| | T1 (No, %) | T2 (No, %) | T1 (No, %) | T2 (No, %) | T1 (No, %) | T2 (No, %) | T1 (No, %) | T2 (No, %) | T1 (No, %) | T2 (No, %) | T1 (No, %) | T2 (No, %) |
| **Internalizing problems scores** | | | | | | | | | | | | |
| Somatic symptoms | 10(10.2) | 5(11.4) | 6(10.3) | 4(6.4) | 29(39.8) | 25(45.5) | 42(55.2) | 37(53.4) | 29(36.7) | 24(34.9) | 34(50.8) | 26(48.5) |
| Worries | 11(8.7) | 9(7.5) | 13(10.3) | 13(11.0) | 38(29.8) | 37(31.4) | 68(53.8) | 64(53.0) | 62(48.9) | 57(47.0) | 61(48.4) | 60(50.1) |
| Unhappy | 7(6.6) | 8(7.4) | 8(9.8) | 7(6.5) | 24(29.5) | 22(29.4) | 42(54.4) | 40(53.7) | 35(47.8) | 33(48.7) | 38(51.8) | 39(54.3) |
| Nervous in new situations | 8(8.2) | 8(9.6) | 10(10.7) | 10(10.5) | 29(31.2) | 27(32.3) | 51(55.5) | 43(53.2) | 41(43.0) | 38(42.9) | 47(51.4) | 45(51.6) |
| Many fears | 7(9.3) | 10(13.7) | 8(7.69) | 9(11.1) | 27(35.38) | 29(39.4) | 41(51.41) | 42(51.9) | 38(50.24) | 34(39.3) | 38(45.96) | 36(44.5) |
| Solitary | 19(9.9) | 12(11.1) | 12(10.7) | 13(12.3) | 39(34.5) | 38(38.3) | 54(48.6) | 46(42.5) | 48(42.7) | 42(38.5) | 60(53.6) | 59(57.3) |
| Has good friend+ | 13 (8.9) | 14(11.0) | 17 (11.7) | 16(13.11) | 47(35.75) | 47(26.6) | 76(44.4) | 71(51.8) | 71 (45.8) | 68(45.6) | 78 (54.5) | 74(51.9) |
| Generally liked+ | 13(9.2) | 12(8.3) | 14(9.7) | 16(11.7) | 44(30.7) | 45(31.5) | 78(48.9) | 73(51.2) | 69(46.0) | 67(47.4) | 77(55.4) | 69(50.0) |
| Picked on or bullied | 9(9.9) | 6(7.9) | 7(10.7) | 6(8.2) | 22(31.2) | 24(43.6) | 33(48.4) | 25(46.0) | 32(50.4) | 19(32.4) | 36(49.5) | 35(61.9) |
| Better with adults | 13(9.42) | 12(8.9) | 15(11.00) | 16(12.0) | 48(33.36) | 47(32.9) | 75(52.13) | 76(53.8) | 67(46.96) | 61(42.8) | 73(47.13) | 74(49.7) |
| **Externalizing problem scores** | | | | | | | | | | | | |
| Tempers | 6(7.1) | 5(9.3) | 4(4.7) | 5(8.1) | 20(29.9) | 23(47.9) | 31(58.7) | 25(50.3) | 23(33.5) | 16(33.3) | 31(65.4) | 24(51.2) |
| Obedient+ | 13(9.2) | 12(7.9) | 17(11.8) | 15(10.7) | 48(35.9) | 42(29.9) | 79(46.9) | 80(51.7) | 72(45.6) | 70(50.4) | 75(50.4) | 74(49.5) |
| Fights or bullies | 6(11.5) | 6(10.7) | 5(9.9) | 6(10.0) | 18(35.7) | 23(49.5) | 20(37.4) | 17(31.0) | 25(48.6) | 20(37.3) | 29(57.0) | 31(61.6) |
| Lies or cheats | 16(11.3) | 6(10.7) | 4(7.4) | 4(7.7) | 17(32.3) | 19(48.4) | 22(42.4) | 23(39.4) | 23(44.6) | 20(20.4) | 31(61.9) | 26(58.7) |
| Steals | 4(12.12) | 3(9.0) | 1(3.23) | 3(9.0) | 12(37.53) | 14(44.6) | 11(34.11) | 9(20.0) | 17(53.86) | 12(37.3) | 19(59.14) | 33(71.8) |
| Restless | 9 (8.9) | 7(7.9) | 8(7.2) | 6(7.9) | 28(30.1) | 28(32.2) | 44 (45.0) | 44(51.0) | 44 (47.2) | 33(41.2) | 54 (61.4) | 51(61.4) |
| Fidgety | 5(5.2) | 8(10.6) | 7(7.3) | 8(9.0) | 18(24.7) | 22(40.5) | 33(54.3) | 28(41.0) | 30(49.1) | 27(43.0) | 36(59.4) | 33(56.0) |
| Easily distracted | 9(8.7) | 7(7.1) | 9(9.4) | 13(11.5) | 28(28.4) | 37(39.0) | 46(50.6) | 47(47.4) | 37(38.3) | 37(39.0) | 61(64.8) | 53(55.8) |
| Thinks before acting+ | 12(8.72) | 12(8.7) | 16(9.68) | 17(11.1) | 49(34.48) | 44(32.3) | 80(52.55) | 75(53.3) | 67(42.55) | 67(48.4) | 77(52.03) | 68(46.0) |
| Good attention+ | 13(9) | 12(10.3) | 15(10.69) | 16(11.4) | 47(31.36) | 47(27.4) | 82(49) | 76(48.7) | 71(52.5) | 67(48.4) | 77(47.11) | 77(53.9) |

T1: baseline or time one assessment

T2: follow-up or time two assessment

No (%): number and percent

+ positive statements which need a reverse score

**Table 4. Longitudinal changes of overall and subscales by severity in March 2020 (T1) and March 2021(T2).**

| Overall mental health problems at T1 | Overall metal health problems at T2 (n, %) | | | | |
|---|---|---|---|---|---|
| | **Normal** | **Borderline** | **Abnormal** | **Total at T1** | **p-value*** |
| **Normal** | 222 (84.7) | 15 (5.7) | 25 (9.5) | 262 | 0.3458 |
| **Positive for SDQ score** | | | | | |
| **Borderline** | 18 (62.1) | 9 (31.0) | 2 (6.9) | 29 | |
| **Abnormal** | 14 (37.84) | 8 (21.62) | 15 (40.54) | 37 | |
| **Total at T2** | 254 (77.4) | 32 (9.76) | 42 (12.8) | 328 | |
| **Internalizing problems in 2020** | Internalizing problems in 2021(n, %) | | | | |
| | Normal | Borderline | Abnormal | Total at T1 | 0.1444 |
| **Normal** | 200 (83.7) | 13 (5.4) | 26 (10.9) | 239 | |
| **Positive for internalizing problems** | | | | | |
| **Borderline** | 22 (64.7) | 10 (29.4) | 2 (5.9) | 34 | |
| **Abnormal** | 31 (59.55) | 8 (20.2) | 16 (20.2) | 55 | |
| **Total at T2** | 253 (77.1) | 31 (9.45) | 44 (13.41) | 328 | |
| **Externalizing problems in 2020 (n, %)** | Externalizing problems in 2021(n, %) | | | | |
| | Normal | Borderline | Abnormal | Total at T1 | 0.0641 |
| **Normal** | 278 (91.2) | 10 (3.28) | 17 (5.57) | 305 | |
| **Positive for externalizing problems** | | | | | |
| **Borderline** | 6 (75.0) | 1 (12.5) | 1 (12.5) | 8 | |
| **Abnormal** | 9 (60.0) | 0 | 6 (40.0) | 15 | |
| **Total at T2** | 293 (89.33) | 11 (3.35) | 24 (7.32) | 328 | |

* Wilcoxon matched-pairs signed-rank test.

overall mental health problems, and 22.6% tested positive at T2 (p = 0.3458). The externalizing dimension's difficulty score also increased, but the difference was insignificant: 7.01% at baseline vs 10.67% at follow-up (p = 0.0441). Internalizing problem scores decreased from 27.13% to 22.87%, but the difference was not statistically significant (p = 0.1444) (Table 4).

However, there was a significant improvement in internalizing SDQ scores in our study cohort at T2 (median (IQR): T1: 7 (6–10) vs T2: 7 (5–9), among females at T2 (median (IQR): T1: 7 (6–10) vs T2: 7 (5–9), p = 0.0300), and among older adolescents at T2 (median (IQR): T1: 7 (5–9) vs T2: 4 (2–8), p = 0.048)) (Table 5).

Females (OR: 0.43, 95% CI: 0.26–0.69), late adolescents (aged 17–20 years) (OR: 0.55, 95% CI: 0.33–0.90), and those from middle-class families (OR: 0.42, 95% CI: 0.23–0.77) had significantly lower odds of developing internalizing mental health problems. Similarly, late adolescents (aged 17–20 years) (OR: 0.17, 95% CI: 0.08–0.32), those with average family wealth (OR: 0.30, 95% CI: 0.13–0.68), and those in higher categories (OR: 0.32, 95% CI: 0.14–0.72) had significantly lower odds of developing externalizing mental health problems. Furthermore, having a history of bullying at school (OR: 2.40, 95% CI: 1.02–5.70) significantly increased the odds of developing externalizing mental health problems (Table 6).

## Discussion

This research aimed to determine the dynamics of in-school adolescents' mental health problems over a year in eastern Ethiopia. Our findings revealed that an increase in the externalizing sub-scale from the baseline to the follow-up assessment increased the SDQ scale's median scores. However, there were no statistical differences in the externalizing issues. Over one year, however, there was a significant improvement in the cohort's internalizing problem scores,

Table 5. Internalizing and externalizing scores amongst matched participants (n = 328) and its subgroups of males (n = 167) and females (n = 161).

| Mental health outcomes | At baseline (T1) | At follow-up (T2) | At baseline (T1) | At follow-up(T2) | p-Value * |
|---|---|---|---|---|---|
| | Mean (± SD) | Mean (± SD) | Median (IQR) | Median (IQR) | |
| **All participants** | | | | | |
| SDQ Internalizing | 7.71 ± 3.04 | 7.28 ± 3.04 | 7 (6–10) | 7 (5–9) | 0.0079 |
| SDQ Externalizing | 4.8 ± 3.32 | 5.01 ± 3.44 | 4 (2–7) | 4 (2–7) | 0.3825 |
| **Males** | | | | | |
| SDQ Internalizing | 7.78 ± 2.93 | 7.52 ± 3.19 | 7 (6–10) | 7 (5–10) | 0.1326 |
| SDQ Externalizing | 4.95 ± 3.40 | 5.29 ± 3.44 | 7 (6–10) | 4 (2–8) | 0.1683 |
| **Females** | | | | | |
| SDQ Internalizing | 7.63 ± 3.16 | 7.03 ± 2.86 | 7 (6–10) | 7 (5–9) | 0.0300 |
| SDQ Externalizing | 4.63 ± 3.24 | 4.71 ± 3.43 | 7 (6–10) | 4 (2–6) | 0.9279 |
| **Young adolescent** | | | | | |
| SDQ Internalizing | 7.94 (2.9) | 7.46 (3.1) | 7 (5–9) | 7 (5–10) | 0.0900 |
| SDQ Externalizing | 4.80 (3.15) | 5.01 (3.6) | 5 (2–7) | 5 (2–7) | 1.00 |
| **Late adolescent** | | | | | |
| SDQ Internalizing | 7.53 (3.14) | 7.11 (2.99) | 7 (5–9) | 4 (2–8) | 0.0498 |
| SDQ Externalizing | 4.79 (3.45) | 5 (3.28) | 4 (2–8) | 7 (5–9) | 0.2752 |

* McNemara's chi$^2$ test.

particularly among females and older adolescents. During the follow-up study, older age, female gender, and family wealth index were associated with a lower likelihood of mental health issues.

Table 6. The association between individual factors and internalizing and externalizing mental health outcomes (random-effects logistic regressions).

| Characteristics | Categories | Internalizing problems | P—values | Externalizing problems | P—values |
|---|---|---|---|---|---|
| **Age** | 14–16 years | 1 | 1 | 1 | 1 |
| | 17–20 years | **0.55 (0.33–0.90)** | **0.020** | **0.17 (0.08–0.32)** | **0.000** |
| **Gender** | Male | 1 | 1 | 1 | 1 |
| | Female | **0.43 (0.26–0.69)** | **0.001** | 0.51 (0.25–1.02) | 0.058 |
| **Any chronic disease** | No | 1 | 1 | 1 | 1 |
| | Yes | 1.67 (0.83–3.20) | 0.151 | 2.23 (0.95–5.20) | 0.063 |
| **Wealth index** | Lowest | 1 | 1 | 1 | 1 |
| | Middle | **0.42 (0.23–0.77)** | **0.005** | **0.30 (0.13-.68)** | **0.004** |
| | Highest | 0.65 (0.36–1.20) | 0.165 | **0.32(0.14–0.72)** | **0.006** |
| *Khat* use | No | 1 | 1 | 1 | 1 |
| | Yes | 1.02 (0.507–1.59) | 0.930 | 1.21 (0.54–2.70) | 0.627 |
| **Alcohol drinking** | No | 1 | 1 | 1 | 1 |
| | Yes | 0.58 (0.24–1.34) | 0.24 | 1.81 (0.67–4.90) | 0.239 |
| **Cigarette smoking** | No | | | | |
| | Yes | 1.9 (0.72–5.01) | 0.190 | 2.0 (0.65–6.1) | 0.223 |
| **Bullying at school** | No | | | | |
| | Yes | 1.32 (0.66–2.64) | 0.428 | **2.4 (1.02–5.70)** | **0.043** |
| **FCV-19S** | Low | 1 | 1 | 1 | 1 |
| | High | 1.34 (0.81–2.25) | 0.256 | 0.63 (0.31–1.27) | 0.201 |
| **Risk of exposure to COVID-19** | No | 1 | 1 | 1 | 1 |
| | Yes | 0.97 (0.47–1.99) | 0.934 | 0.89(0.34–2.35) | 0.822 |

In contrast to prior studies [18, 48], we found that boys had more difficulty scoring in internalizing than girls. In other ways, a study conducted in China in 2021 reported similar results to our findings [49]. Boys' poor social interaction could explain this conclusion and more exposure to risky behaviors, such as *khat* use, tobacco use, alcohol use, excessive screen time, and other forms of social media use commonly observed in the region [50–52].

Our findings indicated that the internalizing problem score unexpectedly decreased during the follow-up assessment. This finding is similar to that in the USA [53], China [54], and Kenya [55] which showed a declining rate of youth mental health problems in a follow-up study. According to our findings, a lower incidence of mental health disorders was associated with the female sex among adolescents during this one year. The reason may be due to protective measures such as staying home during the COVID-19 pandemic, which may increase family time and adolescent relationships. It could be because girls have better relationships with their families, peers, and friends than boys. Their gender-based reactions to problems could explain this disparity [56–58].

Our finding suggested that adolescent mental health problems were associated with older age. Compared to earlier adolescents, teenagers aged ≥17 experienced fewer odds of mental health issues. Previous investigations of individual factors associated with teenage mental health problems [54, 59] found similar results. The correlation may be due to cognitive maturity and coping mechanisms of traumatic life events than young teenagers, such as COVID-19 for instance, or better suited to dealing with difficult full-life situations. People with a higher level of education reported less stress and greater control in their daily lives [60–62].

Individuals in life-threatening situations are more likely to develop mental health issues [63, 64]). Because this study was conducted during the COVID-19 pandemic, we anticipated an increase in mental health problems among these susceptible and more vulnerable groups [65]. We assess the fear of COVID-19 and the risk of infection exposure during the pandemic period. In our study sample, there was no significant relationship between adolescent mental health problems and the consequences of the COVID-19 pandemic. Other studies have found that adolescents who fear their COVID-19 score have more mental health problems. COVID-19 has an association with adolescents' mental health difficulties in research from Turkey [66], North California [67], China [68, 69], and Australia [70, 71]. Fear of the current epidemic has been shown to have the likelihood of developing mental health difficulties like psychological distress, panic disorder, post-traumatic stress symptoms, and significantly moderate to severe depressive symptoms [66, 72, 73].

Fear of developing COVID-19 and contracting the virus may be one of the reasons In contrast, house confinement and accompanying social and physical isolation are all critical risk factors for developing mental health problems [74]. In our study environment, illness perception, community awareness, cultural concerns, health beliefs, contradictory information from all sources, perplexing messages from health professionals, and perspectives of friends and social networks may all play a role in the inverse association [75]. More research with a larger sample size is needed to rule out these surprising findings in our study settings with similar study subjects and conditions to corroborate the findings.

## Strengths and limitations of the study

This sample is valuable because it represents a significant population in the region for which mental health status should understand. However, the generalizability of our findings may be limited since the included data from only three schools. Second, even though a multi-method assessment would be ideal for understanding teenagers' mental health concerns, we collect data using self-report measurements, which may disguise some internalizing and externalizing

behaviors. Third, school settings may exacerbate mental health issues, increasing their incidence. The results have also hampered the fact that the data was self-reported, which could have skewed the results due to numerous biases (e.g., method bias, social desirability bias, memory recall bias). Repeating the current study with more prominent and representative samples from inside and beyond the region is recommended to minimize the mentioned limitation.

## Conclusion

The overall mental health problems among in-school adolescents were high during the study period. The Follow-up evaluations revealed that internalizing problems had improved significantly. The female sex, older age, and the average income index were associated with lower odds of mental health problems. School-based programs promoting prevention, early identification, and rapid intervention are essential. Tailored interventions for reducing adolescent mental health problems should focus on middle-aged and male adolescents from low-income families.

## Supporting information

**S1 Data.**
(DTA)

## Acknowledgments

We want to thank Haramaya University for giving us the approval to conduct this research. Additionally, we would like to thank the Harari Region Education Office for organizing the participants, instructors, and school administrators. We sincerely thank the data collectors for planning and completing the task with care.

## Author Contributions

**Conceptualization:** Gari Hunduma, Yadeta Dessie, Biftu Geda, Tesfaye Assebe Yadeta, Negussie Deyessa.

**Data curation:** Gari Hunduma, Yadeta Dessie, Tesfaye Assebe Yadeta, Negussie Deyessa.

**Formal analysis:** Gari Hunduma, Yadeta Dessie, Biftu Geda, Negussie Deyessa.

**Funding acquisition:** Gari Hunduma, Tesfaye Assebe Yadeta.

**Investigation:** Gari Hunduma, Biftu Geda.

**Methodology:** Gari Hunduma, Yadeta Dessie, Biftu Geda, Tesfaye Assebe Yadeta, Negussie Deyessa.

**Project administration:** Gari Hunduma, Negussie Deyessa.

**Resources:** Gari Hunduma, Yadeta Dessie, Tesfaye Assebe Yadeta.

**Software:** Gari Hunduma, Yadeta Dessie, Biftu Geda, Negussie Deyessa.

**Supervision:** Gari Hunduma, Yadeta Dessie, Biftu Geda, Tesfaye Assebe Yadeta, Negussie Deyessa.

**Validation:** Gari Hunduma, Yadeta Dessie, Negussie Deyessa.

**Visualization:** Gari Hunduma.

**Writing – original draft:** Gari Hunduma, Yadeta Dessie, Biftu Geda, Tesfaye Assebe Yadeta, Negussie Deyessa.

**Writing – review & editing:** Gari Hunduma, Yadeta Dessie, Biftu Geda, Tesfaye Assebe Yadeta, Negussie Deyessa.

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
