## [Decision Letter · Decision Letter 0]

6 Jan 2023

PONE-D-22-31044Mental health dynamics of adolescents: A one-year longitudinal study in Harar, eastern Ethiopia.PLOS ONE

Dear Dr. Hunduma,

Thank you for submitting your manuscript to PLOS ONE. After careful consideration, we feel that it has merit but does not fully meet PLOS ONE’s publication criteria as it currently stands. Therefore, we invite you to submit a revised version of the manuscript that addresses the points raised during the review process.

We look forward to receiving your revised manuscript.

Kind regards,

Anthony A. Olashore, MBCHB, FWACP

Academic Editor

PLOS ONE

Journal Requirements:

"6724"

Additional Editor Comments:

I Agree with the first reviewer; the manuscript is difficult to follow due to extensive grammatical errors. Therefore, I suggest you rewrite the manuscript with the help of a native English speaker. 

Reviewers' comments:

Reviewer's Responses to Questions

**Comments to the Author**

1. Is the manuscript technically sound, and do the data support the conclusions?

Reviewer #1: Partly

Reviewer #2: Yes

2. Has the statistical analysis been performed appropriately and rigorously? 

Reviewer #1: Yes

Reviewer #2: Yes

3. Have the authors made all data underlying the findings in their manuscript fully available?

Reviewer #1: No

Reviewer #2: Yes

4. Is the manuscript presented in an intelligible fashion and written in standard English?

Reviewer #1: No

Reviewer #2: Yes

5. Review Comments to the Author

Reviewer #1: The paper is a good start and a good area of interest. But a coherent flow in some areas is lacking, especially the introduction and discussion. I suggest that the authors re-write the manuscript with the help of a native English speaker.

Reviewer #2: The research is important work which is dire of literature in Africa. The work, though the findings are not statistical significant will serve as a pilot for future larger studies. The authors need to make a few corrections i.e. line 34: 'McNemar's test' instead of 'McNamara's test'. line 69: The article you reference states ''Global prevalence of CMD in adolescents was 25.0% and 31.0%, using the GHQ cut-off point of 4 and 3, respectively. '' The female and male figures are not in the paper. both figures are less then 25 so the sentence does not make sense. line 112: Attrition 358 to 328 - Please provide possible reason and its effects on your result.

6. PLOS authors have the option to publish the peer review history of their article (what does this mean?). If published, this will include your full peer review and any attached files.

Reviewer #1: No

Reviewer #2: No

---

## [Author Response · Author response to Decision Letter 0]

17 Feb 2023

We wrote a rebuttal letter addressing each point raised by the academic editor and reviewer(s). we uploaded this letter as a separate file labelled 'Response to Reviewers'. 'Manuscriptr' and 'Revised Manuscript with Track Changes'.

---

## [Decision Letter · Decision Letter 1]

11 Aug 2023

PONE-D-22-31044R1Mental health dynamics of adolescents: A one-year longitudinal study in Harar, eastern Ethiopia.

PLOS ONE

Dear Dr. Hunduma

Thank you for submitting your manuscript to PLOS ONE. The reviewers have accepted your manuscript, but I would recommend making some minor edits to your manuscript for it to be accepted for publication.

Editor's comments

1. Your abstract is too lengthy; try summarising it into 250 words or less.

2. Remove the statement ‘We assumed that the number of in-school adolescents with mental

health problems would remain stable throughout the study’ from the abstract.

3. Can you please clarify what you meant by the statement below?

‘To lessen adolescent mental health issues, quick interventions that emphasize middle age, the male gender, and wealth disparity are required.’

Kindly reword this statement …may be something like ‘interventions should focus on middle-aged and male adolescents from low-income families’ it sounds more reasonable.

4. Some parts of the manuscript contain punctuations before and after in-text citations, e.g., lines 82, 88, and 98; kindly edit these typos. Also, check the spacing between the in-text citations and the last words. In some, there is space between them, while in others none, e.g., lines 69, 81, etc. These are all over the manuscript.

5. Line 296 and in other parts of the manuscripts consider replacing ‘khat chewing, tobacco smoking, alcohol drinking’ with ‘the use of khat, tobacco, and alcohol.’ You can even say Khat use, tobacco use, and alcohol use.

6. End your introduction by explaining how your findings could benefit society, such as through policy changes or improving respondents' health.

7. Last statement in your conclusion on page 22, line 348, remove tolerated.

Please address these concerns before I reach a decision about your manuscript.

We look forward to receiving your revised manuscript.

Kind regards,

Anthony A. Olashore, MBCHB, PhD, FWACP

Academic Editor

PLOS ONE

Journal Requirements:

Additional Editor Comments: Kindly address the comments listed above and resubmit.

**Comments to the Author**

1. If the authors have adequately addressed your comments raised in a previous round of review and you feel that this manuscript is now acceptable for publication, you may indicate that here to bypass the “Comments to the Author” section, enter your conflict of interest statement in the “Confidential to Editor” section, and submit your "Accept" recommendation.

Reviewer #2: All comments have been addressed

Reviewer #3: All comments have been addressed

2. Is the manuscript technically sound, and do the data support the conclusions?

Reviewer #2: Yes

Reviewer #3: Yes

3. Has the statistical analysis been performed appropriately and rigorously? 

Reviewer #2: Yes

Reviewer #3: Yes

4. Have the authors made all data underlying the findings in their manuscript fully available?

Reviewer #2: Yes

Reviewer #3: Yes

5. Is the manuscript presented in an intelligible fashion and written in standard English?

Reviewer #2: Yes

Reviewer #3: Yes

6. Review Comments to the Author

Reviewer #2: Line 123 - 'Harare’s' SHOULD READ 'Harari's'

The document reads well. I am happy for it to be accepted for publication.

Reviewer #3: The research is scientifically and socially justified and the authors clearly highlighted the gap in literature that merited the research. The study methodology was appropriate and data appropriately collected and analyzed. The results were well described and the discussion showed the contribution of the study to existing literature. The limitations of the study were reported and they were acceptable. The references cited were appropriate.

7. PLOS authors have the option to publish the peer review history of their article (what does this mean?). If published, this will include your full peer review and any attached files.

Reviewer #2: No

Reviewer #3: **Yes: **Dr. Radiance Ogundipe

---

## [Author Response · Author response to Decision Letter 1]

18 Aug 2023

We have seen that the comments and suggestions offered by the editor have immensely helped us to improve our manuscript. We have considered every comment raised and have responded point by point, indicating how we addressed them and tracking the changes we have made. The changes are presented in the revised manuscript and also our responses to the comments are provided in the table attached with suggested modification of the authors.

---

## [Decision Letter · Decision Letter 2]

9 Jan 2024

PONE-D-22-31044R2Mental health dynamics of adolescents: A one-year longitudinal study in Harar, eastern Ethiopia.PLOS ONE

Dear Dr. Hunduma,

Thank you for submitting your manuscript to PLOS ONE. After careful consideration, we feel that it has merit but does not fully meet PLOS ONE’s publication criteria as it currently stands. Therefore, we invite you to submit a revised version of the manuscript that addresses the points raised during the review process.

**ACADEMIC EDITOR: **None

Please submit your revised manuscript by Feb 23 2024 11:59PM. If you will need more time than this to complete your revisions, please reply to this message or contact the journal office at plosone@plos.org. Please include the following items when submitting your revised manuscript:A rebuttal letter that responds to each point raised by the academic editor and reviewer(s). You should upload this letter as a separate file labeled 'Response to Reviewers'.A marked-up copy of your manuscript that highlights changes made to the original version. You should upload this as a separate file labeled 'Revised Manuscript with Track Changes'.An unmarked version of your revised paper without tracked changes. You should upload this as a separate file labeled 'Manuscript'.If applicable, we recommend that you deposit your laboratory protocols in protocols.io to enhance the reproducibility of your results. Protocols.io assigns your protocol its own identifier (DOI) so that it can be cited independently in the future. For instructions see: https://journals.plos.org/plosone/s/submission-guidelines#loc-laboratory-protocols. Additionally, PLOS ONE offers an option for publishing peer-reviewed Lab Protocol articles, which describe protocols hosted on protocols.io. Read more information on sharing protocols at https://plos.org/protocols?utm_medium=editorial-email&utm_source=authorletters&utm_campaign=protocols.

We look forward to receiving your revised manuscript.

Kind regards,

Anthony A. Olashore, MBCHB, PhD, FWACP

Academic Editor

PLOS ONE

Journal Requirements:

Reviewers' comments:

Reviewer's Responses to Questions

**Comments to the Author**

1. If the authors have adequately addressed your comments raised in a previous round of review and you feel that this manuscript is now acceptable for publication, you may indicate that here to bypass the “Comments to the Author” section, enter your conflict of interest statement in the “Confidential to Editor” section, and submit your "Accept" recommendation.

Reviewer #4: All comments have been addressed

Reviewer #5: (No Response)

2. Is the manuscript technically sound, and do the data support the conclusions?

Reviewer #4: Yes

Reviewer #5: Partly

3. Has the statistical analysis been performed appropriately and rigorously? 

Reviewer #4: I Don't Know

Reviewer #5: No

4. Have the authors made all data underlying the findings in their manuscript fully available?

Reviewer #4: Yes

Reviewer #5: Yes

5. Is the manuscript presented in an intelligible fashion and written in standard English?

Reviewer #4: Yes

Reviewer #5: Yes

6. Review Comments to the Author

Reviewer #4: The manuscript has been revised accordingly, and now from my side it is acceptable. The statistical part should be revised by an expert

Reviewer #5: 1. This is a good paper. However, it needs some improvements.

2. The abstract should be re-write after attending to recommendations.

2. Authors wrote a very good introduction. However, there is a need to write done hypothesis as authors wanted to test whether there were differences in SDQ score of T1 and T2.

3. Methods:

-Study setting, designs, and samples:

Line 107: how many schools exist in the region and how authors conducted the random selection? Were they choosing one per cluster, or they listed all schools and use MS Excel or any software for random selection...

Same for selection of participants (line 117).

-Data collection tool:

How did authors do to identify participants after one year for T2 data collection and reconcile with T1 data for the sample selection was dependent T1 and T2: did they use identifiable such names, or ID...

Line 130 and 131: authors state that "...we conducted pretests and confirmed Cronbach's alpha for reliability and validity..." Was this conducted as a pilot study? They included how many participants in the pilot study?

-Variables and Measurements: some variables are in the results section in table 1 but authors did not define them in the methods section e.i. Wealth index, Alcohol use, FCV-19S...

-Regression models are parametric statistic tests. This means that to feed variables in the model, one needs to find out, after univariate analysis, about the homoscedasticity and lineality of these variables, not only about a p value of <0.1 or 0.2.

-Authors repeatedly assess Cronbach alpha while this was not part of the study objectives. Yet they did not even define how they interpreted it in the methods section.

4. Results

-Table 3: It could be better if authors calculated the weighted rate.

7. PLOS authors have the option to publish the peer review history of their article (what does this mean?). If published, this will include your full peer review and any attached files.

Reviewer #4: **Yes: **PANCHANAN ACHARJEE

Reviewer #5: **Yes: **Tshitenge, Stephane

---

## [Author Response · Author response to Decision Letter 2]

19 Jan 2024

Author's Responses to Questions or Responses to reviewers 

Review Comments to the Author

Reviewer # 4: 

All comments have been addressed. The manuscript has been revised accordingly, and now from my side it is acceptable. The statistical part should be revised by an expert

Response 

Thank you for your positive feedback and accepted as presented.

To provide a detailed explanation of the statistical methods used in our study, we start by describing the purpose of each test and how it was applied to our data. Wilcoxon matched-pairs signed-rank test: This test is used to compare two related samples, such as before-and-after measurements or matched pairs of subjects. It tests whether the median difference between the two samples is significantly different from zero. In our study, this test was used to compare the median score severity charges of mental health problems among different groups of study participants as depicted in table 4.

• We used McNemar’s Chi-squared test to compare two paired proportions, such as the proportion of subjects who respond positively during the base line assessment and during the follow up assessment. It tests whether the proportion of subjects who change from one category to another is significantly different from what would be expected by chance. We used this test to compare the severity charges of mental health problems among different groups of study participants (table 5).

• Random-effects logistic regressions on panel data: This method is used to analyze longitudinal data, where the same subjects are measured repeatedly over time. It models the relationship between a binary outcome variable (such as the presence or absence of a mental health disorder) and one or more predictor variables (such as age, gender, or treatment group). In our study, this method was used to find the potential factors associated with individual mental health disorders (table 6).

Review Comments to the Author

Reviewer #5: 

1. This is a good paper. However, it needs some improvements.

Response: Thank you for your positive feedback.we try to address the raised issues points by point and we hope that we improved our revised manuscript. 

2. The abstract should be re-write after attending to recommendations.

Response: We appreciate your point. The abstract makes this clear under the conclusion part. We re-write the paragraph in this way. “The prevalence of mental health problems were high among the study cohort. The proportion of overall problems and externalizing problems has increased over time, indicating a deterioration in the mental health of the study cohort. However, the decrease in internalizing problems among older adolescents, girls, and those with an average wealth index is a positive sign. The findings highlight that tailored interventions are required to reduce externalizing problems and maintain the decrease in internalizing problems. These interventions should target middle-aged and male adolescents from low-income families”. 

3. Authors wrote a very good introduction. However, there is a need to write done hypothesis as authors wanted to test whether there were differences in SDQ score of T1 and T2.

Response: We are grateful and the comment is acceptable. We hypothesized that the number of in-school adolescents with mental health problems would remain stable throughout the study, and therefore there would be no significant difference between the SDQ score of T1 and T2.this is included in the revised manuscript. Please kindly refer lines 104-106. 

4. Methods: -Study setting, designs, and samples: Line 107: how many schools exist in the region and how authors conducted the random selection? Were they choosing one per cluster, or they listed all schools and use MS Excel or any software for random selection... Same for selection of participants (line 117).

Response: Thank you for raising this point. During the data collection period, seven governmental high schools were in the region. We used a simple random sampling technique to select the schools. We listed all schools and randomly selected three schools using computer-generated sampling. We identified grade levels for each selected school, and finally, sections were selected proportionally using the lottery method from each grade level, considering the number of sections. Finally, participants were selected randomly from each section using systematic random sampling. All are incorporated in the revised manuscript. 

5. Data collection tool: How did authors do to identify participants after one year for T2 data collection and reconcile with T1 data for the sample selection was dependent T1 and T2: did they use identifiable such names, or ID...

Response: Thank you for bringing up such crucial points. In order to ensure that the data collected from participants at different times could be linked, we used a unique ID code to identify participants for T2 data collection after a year. This allowed us to reconcile the data with T1 data analysis. It is included in the revised manuscript. 

6. Line 130 and 131: authors state that "...we conducted pretests and confirmed Cronbach's alpha for reliability and validity..." Was this conducted as a pilot study? They included how many participants in the pilot study?

Response: Thank you for your feedback. We conducted pretests on 5% of the sample size to confirm the reliability and validity of the items. All data collection instruments were pre-tested in Dire Dawa administrative counselling among similar in-school adolescents in 5% of the sample size. However, we did not conduct this as a pilot study. We calculated Cronbach’s alpha for the reliability and validity of the tool to assess the scales’ internal consistency and reliability before actual data collection. We apologize for any confusion this may have caused. And it is incorporated the revised manuscript as shown in track change. 

7. Variables and Measurements: some variables are in the results section in table 1 but authors did not define them in the methods section e.i. Wealth index, Alcohol use, FCV-19S...

Response: We appreciated your comments. All important variables in the results section are defined in the methods section including Wealth index, substance use, such as alcohol, cigarettes, khat, or other illicit drugs, Bullying at school and Self-esteem:as per the suggestions. 

8. Regression models are parametric statistic tests. This means that to feed variables in the model, one needs to find out, after univariate analysis, about the homoscedasticity and lineality of these variables, not only about a p value of <0.1 or 0.2.

Response: Thank you for your feedback. We have checked the assumptions of linearity, independence, homoscedasticity, and normality of the model during our data analysis. We performed univariate analysis to check for the homoscedasticity and linearity of the variables before feeding them into the model. We also checked for normality by examining the distribution of the residuals. We ensured that these assumptions were met before feeding the variables into the regression model and it is incorporated in the revised manuscript. Thank you for bringing this to my attention.

9. Authors repeatedly assess Cronbach alpha while this was not part of the study objectives. Yet they did not even define how they interpreted it in the methods section.

Response: Thank you for your feedback. We acknowledge that Cronbach’s alpha was assessed repeatedly in our study. We will ensure that we only assess Cronbach’s alpha if it is relevant to the study objectives in future studies. Additionally, we defined how Cronbach’s alpha was interpreted in the methods section of the study to help readers understand the rationale behind the assessment of Cronbach’s alpha. It is described as “Cronbach’s alpha is a measure of the internal consistency or reliability of a set of survey items. It quantifies the level of agreement on a standardized 0 to 1 scale, with higher values indicating higher agreement between items. Cronbach’s alpha is used to determine whether a collection of items consistently measures the same characteristic. In this study, we assessed Cronbach’s alpha to evaluate the internal consistency of our survey items” in the revised manuscript. 

10. Results -Table 3: It could be better if authors calculated the weighted rate.

Response: Thank you for your feedback. Our intention was to show the prevalence distributions of self-reported adolescents’ mental health problem symptoms with ages and the corresponding time of the survey (T1 & T2). We agree that calculating the weighted rate could be useful in adjusting for differences in the distribution of confounding variables between the study groups. However, we believe that our study groups were similar in terms of these variables, and therefore, calculating the weighted rate may not be necessary. We hope this explanation helps clarify our reasoning for not calculating the weighted rate.”

---

## [Decision Letter · Decision Letter 3]

5 Mar 2024

Mental health dynamics of adolescents: A one-year longitudinal study in Harar, eastern Ethiopia.

PONE-D-22-31044R3

Dear Dr. Hunduma,

We’re pleased to inform you that your manuscript has been judged scientifically suitable for publication and will be formally accepted for publication once it meets all outstanding technical requirements.

Kind regards,

Anthony A. Olashore, MBCHB, PhD, FWACP

Academic Editor

PLOS ONE

**Comments to the Author**

1. If the authors have adequately addressed your comments raised in a previous round of review and you feel that this manuscript is now acceptable for publication, you may indicate that here to bypass the “Comments to the Author” section, enter your conflict of interest statement in the “Confidential to Editor” section, and submit your "Accept" recommendation.

Reviewer #5: All comments have been addressed

2. Is the manuscript technically sound, and do the data support the conclusions?

Reviewer #5: Yes

3. Has the statistical analysis been performed appropriately and rigorously? 

Reviewer #5: Yes

4. Have the authors made all data underlying the findings in their manuscript fully available?

Reviewer #5: Yes

5. Is the manuscript presented in an intelligible fashion and written in standard English?

Reviewer #5: Yes

6. Review Comments to the Author

Reviewer #5: Although a few recommendations are not addressed, most are. The paper can be processed for the next step.

7. PLOS authors have the option to publish the peer review history of their article (what does this mean?). If published, this will include your full peer review and any attached files.

Reviewer #5: No

---

## [Editor Report · Acceptance letter]

21 Mar 2024

PONE-D-22-31044R3 

PLOS ONE

Dear Dr. Hunduma, 

I'm pleased to inform you that your manuscript has been deemed suitable for publication in PLOS ONE. Congratulations! Your manuscript is now being handed over to our production team.

Kind regards, 

on behalf of

Dr. Anthony A. Olashore 

Academic Editor

PLOS ONE